# Understanding the treatment burden of people with chronic conditions in Kenya: A cross-sectional analysis using the Patient Experience with Treatment and Self-Management (PETS) questionnaire

Hillary Koros[1]☯, Ellen Nolte[2]☯*, Jemima Kamano[3], Richard Mugo[1], Adrianna Murphy[2], Violet Naanyu[1,4], Ruth Willis[2], Triantafyllos Pliakas[5], David T. Eton[6], Edwine Barasa[7,8], Pablo Perel[9]

1 Academic Model Providing Access to Health Care, Eldoret, Kenya, 2 Department of Health Services Research and Policy, London School of Hygiene & Tropical Medicine, London, United Kingdom, 3 School of Medicine, Moi University, Eldoret, Kenya, 4 School of Arts and Social Sciences, Moi University, Eldoret, Kenya, 5 Department of Public Health, Environments and Society, London School of Hygiene & Tropical Medicine, London, United Kingdom, 6 Division of Health Care Delivery Research, Robert D. and Patricia E. Kern Center for the Science of Health Care Delivery, Mayo Clinic, Rochester, Minnesota, United States of America, 7 Health Economics Research Unit, KEMRI Wellcome Trust Research Programme, Nairobi, Kenya, 8 Nuffield Department of Medicine, Centre for Tropical Medicine and Global Health, University of Oxford, Oxford, United Kingdom, 9 Department of Non-Communicable Disease Epidemiology, London School of Hygiene and Tropical Medicine, London, United Kingdom

☯ These authors contributed equally to this work.
* ellen.nolte@lshtm.ac.uk

## Abstract

In Kenya, non-communicable diseases (NCDs) are an increasingly important cause of morbidity and mortality, requiring both better access to health care services and self-care support. Evidence suggests that treatment burdens can negatively affect adherence to treatment and quality of life. In this study, we explored the treatment and self-management burden among people with NCDs in in two counties in Western Kenya. We conducted a cross-sectional survey of people newly diagnosed with diabetes and/or hypertension, using the Patient Experience with Treatment and Self-Management (PETS) instrument. A total of 301 people with diabetes and/or hypertension completed the survey (63% female, mean age = 57 years). They reported the highest treatment burdens in the domains of medical and health care expenses, monitoring health, exhaustion related to self-management, diet and exercise/physical therapy. Treatment burden scores differed by county, age, gender, education, income and number of chronic conditions. Younger respondents (<60 years) reported higher burden for medication side effects (p<0.05), diet (p<0.05), and medical appointments (p = 0.075). Those with no formal education or low income also reported higher burden for diet and for medical expenses. People with health insurance cover reported lower (albeit still comparatively high) burden for medical expenses compared to those without it. Our findings provide important insights for Kenya and similar settings where governments are working to achieve universal health coverage by highlighting the

**Data Availability Statement:** All relevant data are included within the paper and its Supporting Information files.

**Funding:** The study was funded by Medical Research Council, United Kingdom (Grant ID: MR/T023538/1). The Principal Investigator is PP and JK is Co-Principal Investigator. EN, VN, AM and EB are the Co-Investigators. The funders had no role in study design, data collection and analysis, decision to publish, or preparation of the manuscript.

**Competing interests:** The authors have declared that no competing interests exist.

importance of financial protection not only to prevent the economic burden of seeking health care for chronic conditions but also to reduce the associated treatment burden.

## Introduction

The prevalence of chronic conditions is rising globally, and many countries in sub-Saharan Africa are now facing the double burden of infectious and non-communicable diseases (NCDs) [1]. In Kenya, NCDs account for almost a third of all deaths and this proportion is projected to rise by over 50% during the next decade [2]. Cardiovascular diseases and cancers are among the most common causes of morbidity, after infectious diseases, and in 2015, about a quarter of the population had hypertension and 5% had diabetes or impaired fasting glycaemia [2]. Availability of screening, early detection and management of NCDs in primary care settings in Kenya is limited. Although strengthening primary health care is a priority [3], the health system has remained hospital-centric, with long waiting times and reduced quality of care [4], resulting in poor retention in care of people who screen positively for NCDs and low treatment adherence [5,6].

Availability of and access to services are important for the successful management of NCDs. However the social and economic contexts within which people with NCDs live also greatly impact their capacity to seek care and self-manage their conditions [7]. Evidence from Malawi and South Africa has pointed to the struggles people with diabetes and/or hypertension face on a daily basis [8–10]. For example, Matima et al. explored the experiences of people with HIV and diabetes comorbidity using the cumulative complexity model developed by Shippee and colleagues, which emphasises how clinical and social factors accumulate and interact to complicate patient care [9,11]. They identified two sets of workloads people have to deal with: 'clinic-related' workload around lack of service integration and perceived power imbalances between the patient and the health care provider, and 'self-care' related workloads around nutritional needs, medication burden and stigma. Available evidence suggests that a high (perceived) burden resulting from patient workloads may negatively affect adherence to treatment and quality of life, in particular among people with multiple and chronic conditions [12–15]. These challenges are further exacerbated by financial concerns, with households in Kenya facing a significant economic burden associated with NCD diagnosis and treatment costs [16].

Much of the work on patient work and treatment burden has been conducted in high-income countries [17,18] and similar work is only beginning to be undertaken in low- and middle income countries [19]. This study seeks to contribute to this emergent evidence by exploring the treatment and self-management burden among people with chronic conditions, in particular hypertension or diabetes, in Kenya.

## Methods

This study was set in the context of a wider implementation study that sought to understand the impact and scalability of a novel approach to integrate promotive, preventive, and curative care for diabetes, hypertension, cervical and breast cancer at the primary health care level (Primary Health Integrated Care Project for Chronic Conditions, PIC4C) within the Academic Model Providing Access to Healthcare (AMPATH) programme in Western Kenya [20]. PIC4C was launched in 2018 by the Kenyan Ministry of Health in partnership with

AMPATH/Moi, Access Accelerated and the World Bank. The model was piloted in Busia and Trans Nzoia counties in Western Kenya, which formed the location for the present study.

One key component of our study was to understand the treatment and self-management burden among a group of people diagnosed with hypertension and/or diabetes since PIC4C implementation and how the burden is distributed across a range of socio-demographic characteristics. We used the Patient Experience with Treatment and Self-Management (PETS) instrument [21] as it assess treatment burden in patients with chronic health conditions requiring self-management. This comprehensive measure allows understanding what aspects of treatment and self-management prove to be most burdensome to people with chronic conditions in Kenya. The survey served as a basis from which to recruit a subsample of people with hypertension and/or diabetes to further explore treatment burden using in-depth interviews.

The PETS builds on a conceptual measurement framework developed from interviews and discussions with people with multiple chronic conditions in the USA [22] and has since been used in a range of populations with chronic conditions in the USA and Norway, including multiple conditions [14,21], cancer [23,24], heart failure [25], diabetes [26], and hypertension [27]. To our knowledge, this patient-reported measure of treatment burden has not yet been applied in a lower middle-income country.

## Adaptation of the PETS instrument

We used the PETS Questionnaire 60-item version (Vs. 2.0) [28]. It contains 12 multi-item and two single item scales: (1) medical information, (2) medications, (3) medical appointments, (4) monitoring health, (5) diet, (6) exercise or physical therapy, (7) medical equipment, (8) relationships with others, (9) medical and health care expenses, (10) difficulty with health care services, (11) role and social activity limitations, and (12) physical and mental exhaustion. The single-item scales assess respectively bother due to medication side effects and bother due to having to rely on medication. Items are assessed by 4 or 5-point ordinal rating scales (e.g. strongly agree, agree, disagree, strongly disagree or very easy, easy, neither easy nor difficult, difficult, very difficult).

We used a rigorous process to translate the PETS into Swahili, following the FACIT (Functional Assessment of Chronic Illness Therapy) Translation & Linguistic Validation Methodology developed by Eremenco et al. [29] and working with the FACITtrans team to implement the method [30]. This involved seven steps: (1) two (independent) forward translations of the questionnaire from English into Swahili by two native speakers; (2) reconciliation of the two forward translations by a third native speaker; (3) back-translation (blinded) of the reconciled version into English by a native speaker fluent in English; (4) review of back-translation by the FACITtrans team; (5) review and finalisation by a fourth independent native speaker of Swahili; (6) quality review of translations; and (7) formatting and proofreading of the test version by a native speaker (see also S1 Fig). Each step is documented in a separate Word document (available on request). All native speakers involved in the translation of the PETS into Swahili were members of the study team and the wider AMPATH research programme.

In a final step, the translated PETS was tested with six patients with diabetes or hypertension to assess comprehensibility and general relevance of the PETS questions in the Kenyan context, using cognitive interviews. Cognitive interview participants were randomly recruited from outpatients attending the Chronic Disease Management Clinic at Moi Teaching and Referral Hospital (MTRH) during one of the clinic days; they were approached at the point of exiting the clinical consultation or the outpatient department. Interviews took place at MTRH premises, following patient consent, and were conducted in Swahili language (S1 Fig).

Interviews found that the phrasing of PETS items was generally understood, except for the domain medical equipment ('Do you currently use any medical equipment or devices'). Interview participants queried whether the use of medical equipment or devices referred to the clinical setting or to their own homes. To enhance comprehensibility, we amended the question to clarify that this referred to use outside clinics or facilities by adding 'in your own home'. The English language PETS and its final translated version can be made available upon written request to Dr Eton, the principal developer of the measure, at dteton99@gmail.com.

In addition to the PETS, the survey included questions about socio-demographic characteristics (age, ethnic group, marital status, education, work history, household size and income) and medical history, namely whether the respondent had ever been told to have a chronic condition as chosen from a list (e.g., high cholesterol, diabetes, hypertension, myocardial infarction, peripheral artery disease, heart failure) and any medications they are currently taking in relation to the reported condition, location of treatment when unwell, and health insurance (National Hospital Insurance Fund, NHIF) cover.

## Participant recruitment

The overall study was set in Busia and Trans Nzoia counties in Western Kenya, where the PIC4C model of care was implemented across a total of 73 facilities (public dispensaries, health centres and county referral hospitals) [20]. We sought to capture a wide range of people with hypertension and/or diabetes in terms of age, gender and broad socio-demographic status and who were registered with PIC4C facilities. Our sampling strategy was based on random sampling of PIC4C facilities in the two counties. In addition, random sampling was done at each strata, with stratification based on the location (rural/urban) or level (size) and then at the facility-level based on chronic condition, gender and age. We used the ratio for diabetic, hypertensive and people with both diabetes and hypertension of 1:6:2 reflecting recorded prevalence derived from the PIC4C database. We sought to arrive at a total sample of around 300 participants. This sample size was used by the developers of the PETS survey instrument to test its validity [21]. We judged this sample size to be appropriate to assess the feasibility of using a formal instrument (i.e. PETS) to assess the treatment and self-management burden in a group of people identified to have diabetes and/or hypertension, and allow for sub-group analysis by a range of socio-demographic characteristics such as gender, ethnicity, educational attainment and income. Anticipating non-response of 30–45%, we oversampled to around 200 in each county (Busia: 202; Tranz Nzoia: 213) so as to randomly select 150 to be surveyed in each. The study population included all patients who had been screened and had hypertension and/or diabetes confirmed through PIC4C service efforts between 2018 and 2020. We used the permanent and the daily registers held at individual health facilities to identify eligible patients. If the randomly selected patient was not available or did not consent, we moved to the next patient on the list of randomly selected participants. This process was repeated until the required sample in each cluster was achieved.

## Data collection

The survey was interviewer-administered, using the principles of computer-assisted personal interviewing (CAPI) and REDCap [31], a secure web application for building and managing online surveys and databases. Trained research assistants acted as focal persons to reach out to selected survey participants by phone; those who could not be reached by phone call were invited in person. Upon agreement to take part, a date for survey completion was set, seeking to identify dates most convenient for participants. For practical reasons, the survey was conducted in-person at a health facility closest to the invited participants' homes. At the time of

the survey, participants were provided with an information sheet about the study (read out to participants unable to read). Patients were reminded that participation was entirely voluntary and non-participation had no influence on clinical care. Participant consent was sought by means of their signature on two paper copies of the consent form. Participants were reimbursed to cover the cost of travel. Survey data were collect by six trained research assistants between 1 December 2020 and 12 February 2021; the average duration was 60 minutes (allowing for breaks where requested) and the completion rate of PETS items was high (>95%). The only exception was the domain 'medical equipment', which was only completed by 5% (n = 14) survey respondents. We therefore did not consider this domain in the further analysis.

## Analysis

Survey data were analysed using descriptive statistics. Where appropriate, we used independent samples t-tests to compare means, chi square of independence for binary/categorical variables, and one-way ANOVA for variables with more than two groups.

Following Eton et al., we scored all PETS scales in such a way that a higher score indicates greater treatment burden. This means that positively worded items were reverse-coded before scoring. We then generated raw scale scores by summing the unweighted items within each domain. Aggregated subscale scores were prorated for missing data when at least 50% of items were available. This was to account for items that may not be applicable to all respondents [21]. As response scales vary across PETS domains (i.e., there is no single response scale used for all domains) and the number of items is not the same across all of the PETS domain scales, we converted all raw scale scores to a standard 0 to 100 metric to facilitate interpretability, with 0 indicating 'no burden' and 100 indicating 'highest burden.' Higher scores on any PETS domain scale always indicate higher burden. We computed internal consistency reliability for all PETS scales using Cronbach's alpha, with an alpha of $\geq 0.70$ taken to indicate adequate reliability [32]. All analysis were conducted using SPSS Version 28.0.

## Ethical approval

The study received approvals from Moi University Institutional Research and Ethics Committee (FAN:0003586) and the London School of Hygiene & Tropical Medicine (17940) as well as a research permit from the National Commission for Science, Technology and Innovation (NACOSTI/P/20/4880).

## Results

Table 1 presents socio-demographic and health-related characteristics of our patient sample by county. Almost two-thirds of respondents (63%) were female; the mean age was 57 years (range 20–90) and 73% were married. The two county samples differed significantly on several variables, including ethnic group, educational attainment, and current working. For example, while a large proportion identified as Luhya in both counties (57% in Busia and 48% in Trans Nzoia), the second largest group in the Busia sample identified as Teso (33%) whereas in Trans Nzoia sample, about 22% identified as Kalenjin and 19% as Kikuyu. The Busia sample also had a higher proportion of respondents with no or only primary education (78% vs. 60%), and a smaller proportion that reported to be currently working (78% vs. 90%). Among those currently working, over half of the Trans Nzoia sample reported to be self-employed in agriculture (55%) compared to 39% in Busia. Average household income was low in both counties, with about half of respondents reporting a monthly income of less than KShs 3,000 (US$27; £20; €23 in January 2021).

**Table 1.  Socio-demographic and health-related characteristics of the patient sample by county.**

| Sample characteristic | Value or frequency | |
|---|---|---|
| | **Busia (n = 150)** | **Trans Nzoia (n = 151)** |
| Other | 9% (13) | 9% (14) |
| **Marital status, % (N)** | | |
| Married | 72% (108) | 75% (113) |
| Single | 23% (34) | 17% (26) |
| Separated/widowed | 5% (8) | 8% (12) |
| **Educational attainment, % (N)**\*\* | | |
| No formal education | 19% (28) | 11% (17) |
| Primary education | 59% (89) | 49% (74) |
| Secondary/tertiary education | 22% (33) | 40% (60) |
| **Currently working, % (N)**\*\* | | |
| Yes | 78% (117) | 90% (136) |
| No | 22% (33) | 10% (15) |
| **Currently working: Type of work, % (N)**\*\* | | |
| Agriculture | 39% (58) | 55% (83) |
| Self-employed | 25% (38) | 20% (30) |
| Unemployed | 7% (11) | 3% (5) |
| Other | 7% (10) | 12% (18) |
| **Household size, number of HH members, % (N)** | | |
| 1 | 5% (8) | 2% (3) |
| 2 | 8% (12) | 10% (15) |
| 3 | 10% (15) | 11% (17) |
| 4–6 | 43% (54) | 42% (64) |
| 7–9 | 28% (42) | 24% (36) |
| 10 or more | 6% (9) | 11% (16) |
| **Monthly household income past year, KSh, % (N)** | | |
| <1,000 | 25% (37) | 23% (34) |
| 1,000–2,999 | 29% (44) | 27% (41) |
| 3,000–4,999 | 15% (23) | 17% (25) |
| 5,000–7,999 | 11% (17) | 12% (18) |
| 8,000–10,000 | 9% (14) | 7% (10) |
| 10,000–15,000 | 9% (13) | 11% (17) |
| **Self-reported chronic condition, % (N)** | | |
| Hypertension | 85% (127) | 82% (124) |
| Diabetes | 44% (66) | 37% (56) |
| High cholesterol | 5% (7) | 3% (4) |
| HIV | 5% (7) | 2% (3) |
| Stroke | 6% (9) | - |
| Heart disease | 4% (6) | 2% (3) |
| Arthritis | 4% (6) | 1% (1) |
| Ulcer | 2% (3) | 2% (3) |
| Other | 7% (10) | 2% (3) |
| **No. chronic conditions, % (N)**\*\*\* | | |
| 1 | 52% (78) | 71% (107) |
| 2 | 37% (56) | 26% (40) |
| 3 or more | 11% (16) | 3% (4) |
| *Mean number* | *1.6* | *1.3* |

(*Continued*)

**Table 1.** (Continued)

| Sample characteristic | Value or frequency | |
|---|---|---|
| | Busia (n = 150) | Trans Nzoia (n = 151) |
| **Reporting hypertension and diabetes, % (N)** | | |
| Yes | 27% (40) | 21% (32) |
| No | 73% (110) | 79% (119) |
| **Taking any medication (excl. herbal), % (n)** | | |
| Yes | 99% (149) | 100% (151) |
| No | 1% (1) | - |
| **No. of medications taken (excl. herbal), % (N)** | | |
| 1 | 71% (107) | 80% (120) |
| 2 | 27% (41) | 20% (30) |
| 3 | 1% (1) | 1% (1) |
| **Taking herbal drugs, % (N)** | | |
| Yes | 7% (10) | 13% (20) |
| No | 93% (139) | 87% (131) |
| **Location of treatment when unwell, % (N)**\*\* | | |
| County hospital | 10% (15) | 11% (17) |
| Sub-county hosp | 25% (38) | 38% (57) |
| Health centre | 33% (49) | 29% (44) |
| Dispensary | 27% (40) | 18% (27) |
| Private provider | 5% (8) | 4% (6) |
| **Has NHIF cover, % (N)** | | |
| Yes | 24% (36) | 25% (38) |
| No | 76% (114) | 74% (112) |
| Don't know | - | 1% (1) |

Note.

\*\*\* $p < 0.001$ (chi square)

\*\* $p < 0.05$ (chi square).

The majority of respondents reported having one chronic condition, with the Busia sample having a significantly higher proportion of respondents with two or more conditions (48% vs. 30%). Over 80% in each county reported having hypertension and between 37% (Trans Nzoia) and 44% (Busia) had diabetes; the mean number of reported chronic conditions was 1.6 for the Busia sample and 1.3 for the Trans Nzoia sample. Only about a quarter reported having had NHIF cover at the time of the survey in either county.

Table 2 shows reliability of PETS domain scales for the sample overall and by county (S1–S5 Tables show the frequency of responses to individual PETS domain items). Internal consistency reliability was generally good for all multi-item scales, and Cronbach's coefficients were well above the threshold for adequate reliability ($\alpha \geq 0.70$). The only exceptions were 'monitoring health' ($\alpha = 0.55$), 'medical and health care expenses' ($\alpha = 0.69$) and 'diet' ($\alpha = 0.67$), although the latter two were close to the adequate reliability threshold. Monitoring health asks about the ease or difficulty of monitoring health behaviours (e.g. tracking exercise, foods eaten, or medicines taken) and health condition (e.g. weighing, checking blood pressure or blood sugar levels). Our cognitive interviews found that while respondents generally found the question itself easy to understand, some required further explanations of what was meant by 'monitoring'. Furthermore, the ability to track, say, blood pressure or blood sugar levels at home requires respondents to have the relevant equipment at their disposal but, as noted

**Table 2. Reliability (Cronbach's α) of PETS domains scales.**

| PETS scale | Total PIC4C sample | Busia | Trans Nzoia |
|---|---|---|---|
| Medical information (7 items) | 0.83 | 0.92 | 0.87 |
| Medications (7 items) | 0.86 | 0.85 | 0.85 |
| Medical appointments (6 items) | 0.85 | 0.84 | 0.83 |
| Monitoring health (2 items) | 0.55 | 0.64 | 0.39 |
| Interpersonal challenges (4 items) | 0.84 | 0.81 | 0.88 |
| Medical and health care expenses (5 items) | 0.69 | 0.76 | 0.55 |
| Difficulties with health care services (7 items) | 0.82 | 0.56 | 0.95 |
| Role/social activity limitations (6 items) | 0.92 | 0.93 | 0.91 |
| Physical/mental fatigue (5 items) | 0.87 | 0.87 | 0.88 |
| Diet (3 items) | 0.67 | 0.72 | 0.46 |
| Exercise and physical therapy (4 items) | 0.82 | 0.83 | 0.81 |

earlier, only 5% of survey respondents reported using medical equipment that would enable self-monitoring at home.

Lower internal consistency reliability for the domains medical and health care expenses and diet was largely driven by the Trans Nzoia sample (Table 2). The former domain includes one question about the ease or difficulty to understand what is and what is not covered by health insurance. Leaving this item out would increase Cronbach's α for the domain to 0.918 in the Trans Nzoia sample (0.889 in the Busia sample). The item had a relatively large number of 'not applicable' responses (20% in Busia, 25% in Trans Nzoia). As only 25% of sample reported having NHIF cover it is conceivable that the item was not relevant to most respondents, although we have retained it in our subsequent analysis as removing it did not change findings in any discernible way. The domain diet includes the item 'It is hard to find healthy foods', which returned a negative Cronbach's α when deleted for the Trans Nzoia sample, possibly reflecting the limited spread of answers in this sample across three categories ('strongly agree', 'agree', 'disagree') only. Finally, the domain difficulty with health care services showed good internal consistency reliability for the total sample but not for the Busia sample ($α = 0.56$). This might be explained by the relatively large proportion of 'not applicable' responses for three of the items in this domain (S1 Table).

Table 3 presents descriptive statistics of 11 PETS domains and two single item scores for the total sample and by county. The highest treatment burdens were reported in the domains of medical and health care expenses, monitoring health, and physical and mental exhaustion related to self-management as well as diet and exercise/physical therapy. Several items within other domains were also rated as especially burdensome, such as finding transport to get to (S1 Table) and long waits at medical appointments (S2 Table), feeling dependent on others for health care needs (S3 Table) or self-management interfering with work, family responsibilities or daily activities (S4 Table). Fig 1 disaggregates domains with the highest mean burden scores by response item and county.

Compared with the Busia sample, Trans Nzoia mean burden scores were significantly higher for medical information, medications, medical appointments and difficulty with health services. Conversely, Busia respondents reported a significantly higher burden for interpersonal challenges, medical expenses, role/social activity limitations and diet (Table 3). Scale scores were positively skewed toward a lower burden in most domains except for medical expenses, which was slightly negatively skewed toward a higher burden. Diet was also negatively skewed in the Busia sample. Floor effects were generally lower in the Trans Nzoia sample.

**Table 3. Descriptive statistics of 11 PETS domain scale scores and two single-item indicators.**

| PETS domain | N | Mean | SD | Score range | % Floor | % Ceiling | Skewness |
|---|---|---|---|---|---|---|---|
| Medical information (total) | 301 | 33.1 | 21.3 | 0–100 | 9 | 1 | 0.64 |
| Busia | 150 | **29.2** | 25.3 | 0–100 | 18 | 2 | 0.86 |
| Trans Nzoia | 151 | **36.9** | 15.6 | 3.6–75 | <1 | 6.6 | 0.87 |
| p value | | *p = 0.002* | | | | | |
| Medications (total) | 295 | 24.0 | 18.5 | 0–100 | 15 | <1 | 0.99 |
| Busia | 145 | **18.7** | 20.8 | 0–100 | 26.7 | <1 | 1.43 |
| Trans Nzoia | 150 | **29.1** | 14.3 | 0–100 | 2 | <1 | 1.27 |
| p value | | *p<0.001* | | | | | |
| Medical appointments (total) | 201 | 31.8 | 20.2 | 0–91.7 | 10 | <1 | 0.49 |
| Busia | 150 | **27.6** | 23.7 | 0–100 | 17.3 | <1 | 0.74 |
| Trans Nzoia | 151 | **36.0** | 14.9 | 0–75 | 2 | 5.3 | 0.83 |
| p value | | *p<0.001* | | | | | |
| Monitoring health (total) | 296 | 48.9 | 26.6 | 0–100 | 6 | 7 | 0.14 |
| Busia | 150 | 50.9 | 32.0 | 0–100 | 12 | 14 | -0.01 |
| Trans Nzoia | 146 | 46.8 | 19.5 | 0–87.5 | <1 | 3 | 0.21 |
| p value | | *p = 0.187* | | | | | |
| Interpersonal challenges (total) | 301 | 23.9 | 25.9 | 0–93.8 | 36 | 3 | 1.02 |
| Busia | 150 | **29.3** | 29.6 | 0–93.8 | 31 | 5 | 0.74 |
| Trans Nzoia | 151 | **18.4** | 20.4 | 0–98.8 | 40 | <1 | 1.07 |
| p value | | *p<0.001* | | | | | |
| Medical and health care expenses (total) | 301 | 63.0 | 23.0 | 0–100 | 1 | 11 | -0.30 |
| Busia | 150 | **67.3** | 27.2 | 0–100 | 2 | 21 | -0.53 |
| Trans Nzoia | 151 | **58.8** | 17.8 | 0–100 | <1 | <1 | -0.58 |
| p value | | *p = 0.001* | | | | | |
| Difficulty with health care services (total) | 259 | 36.1 | 19.1 | 0–100 | 8 | <1 | 0.30 |
| Busia | 133 | **33.2** | 25.1 | 0–100 | 13 | 1 | 0.53 |
| Trans Nzoia | 126 | **39.0** | 8.5 | 23.8–66.7 | 3 | 1 | 0.88 |
| p value | | *p = 0.015* | | | | | |
| Role/social activity limitations (total) | 301 | 32.1 | 27.5 | 0–100 | 18 | 4 | 0.84 |
| Busia | 150 | **35.3** | 33.9 | 0–100 | 23 | <1 | 0.61 |
| Trans Nzoia | 151 | **28.9** | 18.7 | 0–87.5 | 13 | <1 | 0.57 |
| p value | | *p = 0.043* | | | | | |
| Physical/mental exhaustion (total) | 301 | 39.7 | 21.0 | 0–100 | 5 | 2 | 0.40 |
| Busia | 150 | 39.1 | 25.5 | 0–100 | 7 | 3 | 0.51 |
| Trans Nzoia | 151 | 40.4 | 15.3 | 0–100 | 2 | <1 | -0.01 |
| p value | | *p = 0.593* | | | | | |
| Bother due to reliance on medicine* (total) | 295 | 26.3 | 31.6 | 0–100 | 47 | 9 | 1.05 |
| Busia | 145 | 26.0 | 36.9 | 0–100 | 58 | 14 | 1.07 |
| Trans Nzoia | 150 | 26.5 | 25.6 | 0–100 | 35 | 3 | 0.87 |
| p value | | *p = 0.900* | | | | | |
| Bother due to side effects of medicine* (total) | 295 | 26.7 | 30.3 | 0–100 | 43 | 7 | 0.98 |
| Busia | 145 | 29.0 | 36.2 | 0–100 | 51 | 12 | 0.86 |
| Trans Nzoia | 150 | 24.5 | 23.1 | 0–100 | 35 | 1 | 0.77 |
| p value | | *p = 0.200* | | | | | |
| Diet (total) | 279 | 58.3 | 25.6 | 0–100 | 5 | 10 | -0.23 |
| Busia | 137 | **62.0** | 32.4 | 0–100 | 9 | 18 | -0.54 |
| Trans Nzoia | 142 | **54.6** | 16.1 | 22.2–100 | 1 | <1 | 0.32 |

*(Continued)*

**Table 3.** (Continued)

| PETS domain | N | Mean | SD | Score range | % Floor | % Ceiling | Skewness |
|---|---|---|---|---|---|---|---|
| p value | | p = 0.015 | | | | | |
| Exercise/physical therapy (total) | 270 | 43.1 | 25.8 | 0–100 | 9 | 5 | 0.36 |
| Busia | 130 | 41.7 | 32.9 | 0–100 | 15 | 9 | 0.42 |
| Trans Nzoia | 140 | 44.5 | 16.9 | 0–91.7 | 1 | <1 | 0.30 |
| p value | | p = 0.386 | | | | | |

Note.

* single item scales; numbers in bold: Difference between Busia and Trans Nzoia sample statistically significant.

There were significant differences in mean burden scores by ethnic group in the domains medications, medical appointments and difficulties with health care services, with those identifying as Teso reporting a substantially lower burden. For example, in the domain difficulties with health care services, Teso respondents had a mean score of 21.3 (SD 20.0) compared with a score of around 40 for the other three main ethnic groups (S6 Table). At the same time, Teso respondents reported a significantly higher burden in the relationship with others and medical expenses domains (S6 Table).

There were generally fewer differences in mean burden scores by age, gender, education, income or number of chronic conditions. Younger respondents (<60 years) reported a significantly higher burden for bother with medication side effects and for diet (p < .05); they also reported a higher burden for medical appointments, which was borderline significant (p = 0.075). Respondents with no formal education or those on low income also reported a significantly higher burden for diet as well as for medical expenses (S6 Table). Perhaps not surprisingly, those reporting having health insurance cover reported a significantly lower (albeit still comparatively high) burden for medical expenses compared to those without health insurance cover (53.1 (SD 25.7) vs. 66.3 (SD 21.1).

Fig 2 disaggregates PETS scores for the total sample by number of chronic conditions. Perhaps somewhat counterintuitively, we found higher scores for people reporting one chronic condition compared to those with two or more conditions in several domains although differences were not statistically significant. The only domains where burden scores were significantly higher with more chronic conditions were role/social activity limitations and diet. There was a clearer relationship between the number of drugs taken and reported treatment burden in most domains although differences were significant for bother with medicine reliance, bother with medication side effects, role/social activity limitations and diet only (S6 Table).

Exploring treatment and self-management burden for type of chronic condition, we found that people with diabetes (with or without other conditions) reported a higher burden in all domains compared to people with any other NCD (including hypertension), and a significantly higher burden for medical appointments, role/social activity limitations, bother with medication side effects, and diet (S6 Table).

## Discussion

To our knowledge, this is one of the first studies that have used a patient-reported measure of treatment and self-management burden among people with chronic conditions in a lower middle-income country. We found that patients in Western Kenya who were recently diagnosed with hypertension or diabetes reported a considerable treatment burden in a range of areas, with particularly high burdens around difficulty paying for health care, monitoring

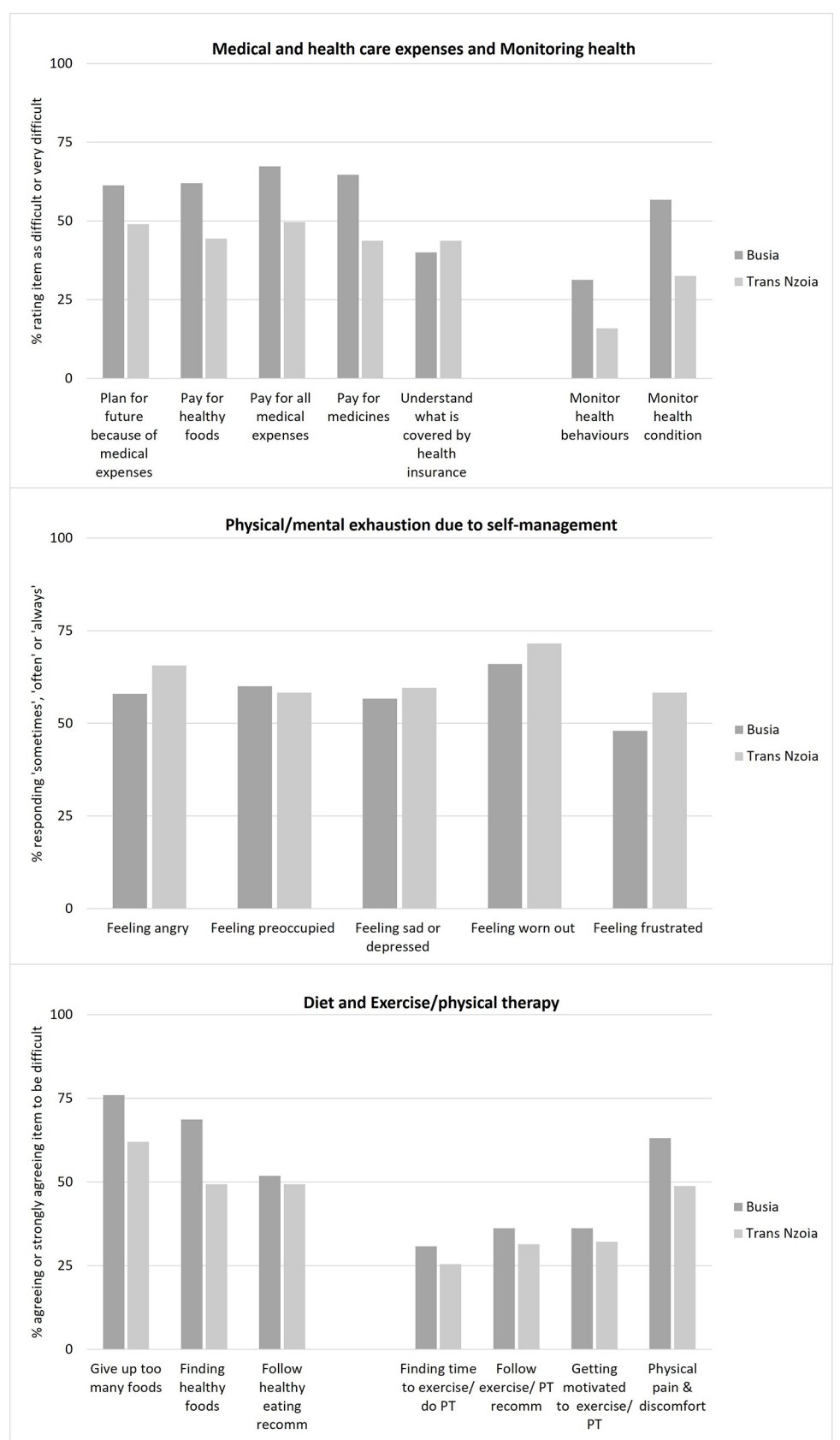

**Fig 1. Response frequency for selected PETS domain items, by county.**

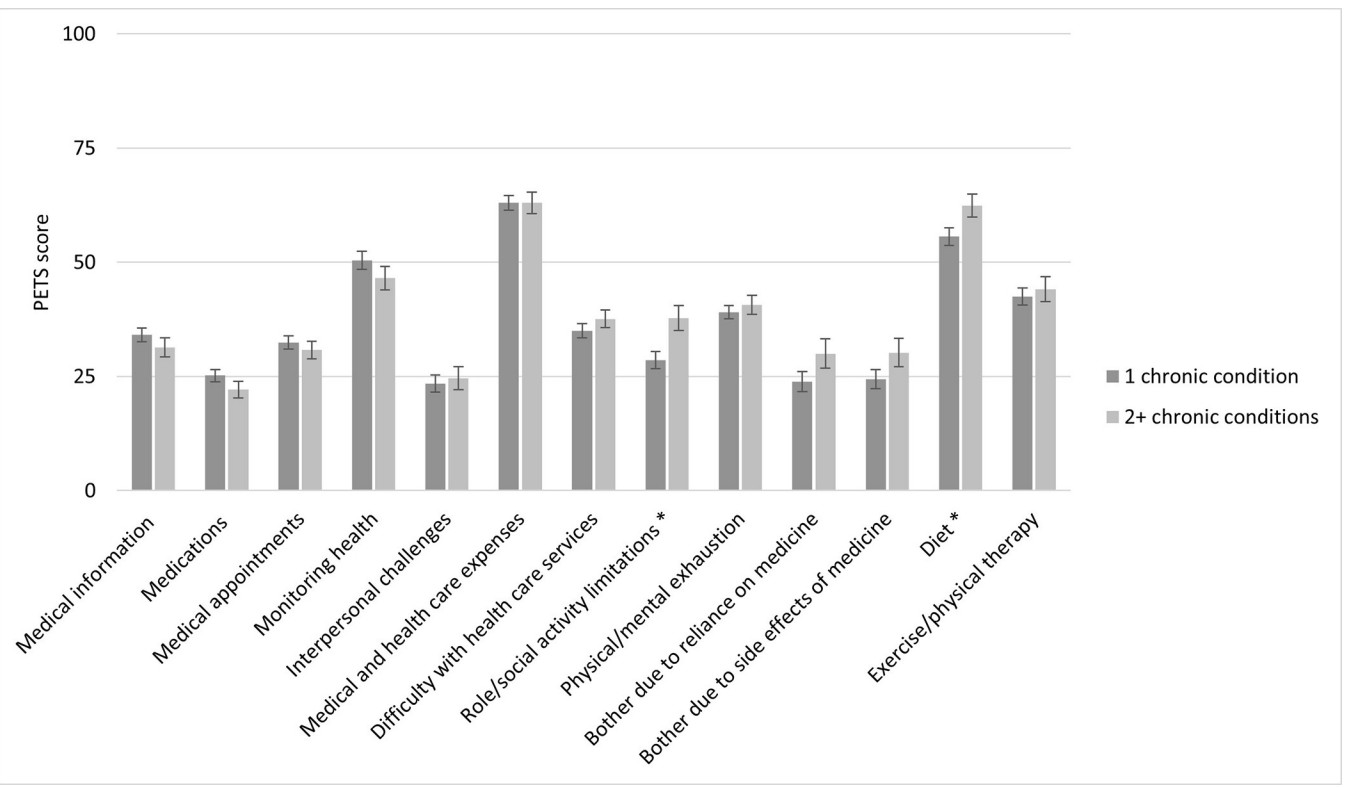

**Fig 2. PETS scores by number of chronic conditions, total sample.** Note. * role/social activity limitations: p = 0.005; diet: p = 0.030. See also S7 Table.

health, and physical or mental exhaustion from self-management, alongside affording or following a healthy diet and engaging with exercise or physical therapy. Other areas perceived as especially difficult or bothersome included finding transport to get to and long waits at medical appointments, feeling dependent on others for health care needs and the impact that self-management had on work, family responsibilities or daily activities.

The reported treatment and self-management burden differed between patient populations in the two counties in most domains, and this appeared to be driven by a combination of factors, including ethnic group, educational level and burden of multiple chronic diseases. However, the group sizes were too small to allow for further robust analysis of underlying patterns.

Empirical application of the PETS has so far only been documented for populations in the USA and Norway, and the reported treatment and self-management burden is consistent with our findings in so far as the highest burden among people with diabetes or multimorbidity was in the domains of medical expenses, monitoring health and physical or mental exhaustion from self-management [21,33]. PETS scores in US patient populations tended to be lower than in the Kenyan samples queried in this study, although in Eton et al.'s 2017 study, a subgroup of participants recruited from an urban safety-net hospital (Hennepin County Medical Center in Minneapolis, Minnesota), which provides care for low-income, uninsured, and vulnerable persons, had mean PETS scores that were much closer to those reported in our sample [21]. An additional study by Eton et al. [34] developed and administered a briefer short-form version of the PETS with the same patient population and found even higher mean scores in several burden domains. However, caution must be exercised when making direct comparisons given the different version of the PETS used in this prior study. Contrary to other measures of treatment burden, such as the Treatment Burden Questionnaire (TBQ) [35] or the Multimorbidity

Treatment Burden Questionnaire (MTBQ) [13], psychometric testing of the PETS has not supported a global summary burden score, although more recent work has distinguished 'workload' and 'impact' summary scores which aggregate some of the PETS domain scales [14]. Determination of severity thresholds for PETS domain scores (i.e., low, medium, and high burden) are currently pending and may make comparisons with scores from other measures more feasible in the future.

Overall, our findings for treatment burden in a population of people with diabetes and/or hypertension in Western Kenya generally align well with other studies that used comparable measures. Work that has assessed treatment burden in different populations found that younger people [13,34,36], those with greater financial difficulties [13,21,35], and those with multiple conditions tended to report higher treatment burdens [14,21], although the international evidence is somewhat mixed on the latter [13,14,35]. We also showed treatment burden to be significantly associated with younger age although this was in selected domains around medical appointments, bother with reliance on medicines and diet only. Duncan et al. argued that a higher burden might reflect role differences, with younger people having to organise medical appointments around work commitments and looking after dependents while perhaps also having different expectations in terms of managing their own health [13]. Similar to studies in the USA, lower income was associated with a higher treatment burden in our sample, in particular around medical expenses [14,21]. Studies of populations in countries with universal health systems did not find such associations [13,35,36]. Moreover, we found that respondents who had health insurance cover reported a significantly lower (albeit still high) treatment burden as it related to medical expenses. Taken together, these observations highlight the importance of financial safety netting not only to protect people from financial risk related to managing chronic conditions but also to lower the associated treatment burden.

We did not find treatment burden to be higher among people with multiple chronic conditions except in the domains bother with medicine reliance, role/social activity limitations and diet. Possible explanations for an apparent lack of association between treatment burden and number of chronic conditions include that those with more than one condition might find it easier to call upon and navigate medical and social support because they are more experienced and 'already in the system' [37], and, possibly, because of the integrated provision of services within PIC4C, which would otherwise have required repeat visits to different clinics. We were unable, in this study, to assess disease severity; we did, however, find a dose-response relationship between the number of drugs taken and treatment burden, which could be indicative of greater perceived or experienced severity. Perhaps not surprisingly, we found that people with diabetes reported a higher treatment burden compared to those with hypertension. While it is difficult to compare directly, studies of people with (multiple) chronic conditions in Switzerland (using the TBQ) [38] and Victoria, Australia (MTBQ) [39] also found a positive association of treatment burden with diabetes. Herzig et al. [38] suggested that a perceived high treatment burden for diabetes might reflect the wider range of activities that patients have to engage with to effectively manage the condition, from regular drug intake to adapting diet and exercise, all impacting on perceived quality of life. Evidence from low resource settings specifically points to the key challenges of affording and accessing a healthy diet among people with diabetes [40,41].

## Strengths and limitations

A key strength of our study was the high completion rate of all items of the PETS instrument (>95%). All items appeared to be relevant for all patients and the proportion of 'does not apply' responses was low for most. One exception were selected items in the 'difficulty with

health care services' domain, where about 35% of the total sample responded not applicable to the first two response items ('different providers not communicating'; 'seeing too many different specialists'). Reliability scores and the overall coherence of our findings in relation to what is known in the Kenyan context and internationally supports the transferability and applicability of the concept of treatment burden, and the use of the PETS in describing it, to Kenya.

Our sample was not representative of the patient population in Busia and Trans Nzoia counties and we cannot, therefore, generalise across the wider patient population in either county. However, this was not the aim of this survey. Indeed, the sample was meant to capture people using PIC4C services with targeted sampling of those with hypertension and/or diabetes specifically. Some two-thirds of our sample were women, which broadly reflects the general pattern of service use across PIC4C facilities, with women more likely to attend care for hypertension or diabetes as recorded in the PIC4C database. The Busia and Trans Nzoia samples differed significantly in terms of ethnic group composition; identification as belonging to a given ethnic group was self-reported although the proportions in our sample appear to reflect the ethnic composition of Busia and Trans Nzoia counties in broad terms. Indeed, differences in ethnic composition was a main reason for selecting the two counties as pilot regions for the wider PIC4C study.

About half of our respondents reported a monthly income of less than KShs 3,000, which is lower than the national poverty line of KShs 3,252 for rural populations as defined by the Kenyan National Bureau of Statistics [42], although similar to poverty levels reported for Busia and Trans Nzoia counties, at 61% and 50%, respectively (2018) [20]. However, it is important to note that household income data were self-reported and not comprehensively captured by the survey; more than half of respondents were farmers (56%), whose household income is difficult to estimate with considerable monthly fluctuation. Household expenditure is considered a preferable measure in settings characterised by mostly informal economic activities and income cannot easily be tracked or quantified [43]; however, this was not possible in the context of this study.

## Implications for practice and research

A key observation of our study is that people with diabetes and/or hypertension in Western Kenya reported a high treatment burden in a range of domains. While the substantial economic burden of chronic illness faced by individuals and households in Kenya is well known [44], the burden resulting from monitoring health and the physical/mental burden from self-management have as yet not been documented. The further development of integrated chronic care programmes such as PIC4C and similar programmes elsewhere should make provisions for supporting people to alleviate the added burdens in order to optimise NCD management and, ultimately, outcomes.

Future programmes should also consider targeting specific groups with higher burdens specifically. These include for example younger patients with dependents who have to balance work and caring commitments alongside managing their health condition/s. Effective management will require long-term engagement over the life time, which younger people may find especially challenging and they might benefit from targeted practical support.

Similarly, the main areas reported to be especially bothersome were finding transport to get to and long waits at medical appointments. These are areas where targeted approaches can potentially make a substantial difference to people, through for example, outreach services such as group medical visits as previously trialled in Western Kenya [45,46]. Recent efforts in the study region saw the piloting of tele-medicine services using community health workers and peer support as 'clinician-extenders' during the COVID-19 pandemic to maintain and

improve access to NCD care [47]. Such approaches provide a useful starting point for the further development of NCD programmes in the region.

Finally, our observations provide important insights for Kenya as a whole as the government moves to roll out universal health coverage (UHC) [48]. In doing so, there is particular need for providing comprehensive coverage for NCDs that also involves enhanced support for monitoring and self-management to ensure reduced treatment burden.

The PETS has proved to be a useful tool for assessing the treatment and self-management burdens of people with NCDs in Western Kenya at one point in time. Further work should test the instrument in a wider range of populations in different settings and over time to understand its value as a measure of impact of interventions seeking to support people with chronic conditions.

## Supporting information

**S1 Fig. Translation and testing of the PETS using the FACIT translation and linguistic validation methodology.**
(DOCX)

**S1 Table. Frequency of PETS domain items, total sample and by county: Medical information, medication, medical appointments, monitoring health and medical and health care expenses.**
(DOCX)

**S2 Table. Mean score and frequency of PETS domain items: Diet, exercise and physical therapy, and difficulty with health services.**
(DOCX)

**S3 Table. Mean score and frequency of PETS domain items: Reliance on medicine, side effects of medicine, relationship with others.**
(DOCX)

**S4 Table. Mean score and frequency of PETS domain items: Role/social activity limitations due to self-management.**
(DOCX)

**S5 Table. Mean score and frequency of PETS domain items: Physical/mental exhaustion.**
(DOCX)

**S6 Table. Mean PETS domain scores by socio-demographic and health-related characteristics.**
(DOCX)

**S7 Table. PETS scores by number of chronic conditions, total sample.**
(DOCX)

## Acknowledgments

The authors wish to express their gratitude to all patients who have so generously given their time to complete the survey and to Alex Njumwah, Elizabeth Khisa, Kenneth Rotich, Kevinah Asigi, Ruth Nehema, Shameem Mutalib and Topista Nafula, who have conducted the fieldwork.

## Author Contributions

**Conceptualization:** Ellen Nolte, Jemima Kamano, Adrianna Murphy, Violet Naanyu, Edwine Barasa, Pablo Perel.

**Data curation:** Ellen Nolte.

**Formal analysis:** Hillary Koros, Ellen Nolte, Triantafyllos Pliakas.

**Funding acquisition:** Ellen Nolte, Jemima Kamano, Adrianna Murphy, Violet Naanyu, Edwine Barasa, Pablo Perel.

**Investigation:** Hillary Koros, Richard Mugo.

**Methodology:** Hillary Koros, Ellen Nolte, Adrianna Murphy, Violet Naanyu, David T. Eton.

**Project administration:** Hillary Koros, Jemima Kamano, Richard Mugo.

**Resources:** Jemima Kamano, Richard Mugo.

**Software:** Richard Mugo.

**Supervision:** Ellen Nolte, Jemima Kamano, Violet Naanyu, Pablo Perel.

**Validation:** Ellen Nolte, Jemima Kamano, Richard Mugo, Ruth Willis, David T. Eton, Pablo Perel.

**Visualization:** Ellen Nolte.

**Writing – original draft:** Ellen Nolte.

**Writing – review & editing:** Hillary Koros, Ellen Nolte, Jemima Kamano, Richard Mugo, Adrianna Murphy, Violet Naanyu, Ruth Willis, Triantafyllos Pliakas, David T. Eton, Edwine Barasa, Pablo Perel.

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
