## [Decision Letter · Decision Letter 0]

7 Oct 2022

PGPH-D-22-01437

Understanding the treatment burden of people with chronic conditions in Kenya: a cross-sectional analysis using the Patient Experience with Treatment and Self-Management (PETS) questionnaire

Dear Dr. Nolte,

Thank you for submitting your manuscript to PLOS Global Public Health. After careful consideration, we feel that it has merit but does not fully meet PLOS Global Public Health’s publication criteria as it currently stands. Therefore, we invite you to submit a revised version of the manuscript that addresses the points raised during the review process.

Kindly address the reviewers' comments specific to the validity of the tool used for measurement. The validation part is not described in detail and may merit a separate publication elsewhere. 

We look forward to receiving your revised manuscript.

Kind regards,

Panniyammakal Jeemon

Academic Editor

Journal Requirements:

Additional Editor Comments (if provided):

Reviewers' comments:

Reviewer's Responses to Questions

**Comments to the Author**

1. Does this manuscript meet PLOS Global Public Health’s publication criteria? Is the manuscript technically sound, and do the data support the conclusions? The manuscript must describe methodologically and ethically rigorous research with conclusions that are appropriately drawn based on the data presented.

Reviewer #1: Partly

Reviewer #2: Yes

Reviewer #3: Yes

Reviewer #4: Partly

Reviewer #5: Yes

2. Has the statistical analysis been performed appropriately and rigorously?

Reviewer #1: Yes

Reviewer #2: Yes

Reviewer #3: Yes

Reviewer #4: Yes

Reviewer #5: Yes

3. Have the authors made all data underlying the findings in their manuscript fully available (please refer to the Data Availability Statement at the start of the manuscript PDF file)?

Reviewer #1: No

Reviewer #2: Yes

Reviewer #3: Yes

Reviewer #4: Yes

Reviewer #5: Yes

4. Is the manuscript presented in an intelligible fashion and written in standard English?

Reviewer #1: Yes

Reviewer #2: Yes

Reviewer #3: Yes

Reviewer #4: Yes

Reviewer #5: Yes

5. Review Comments to the Author

Reviewer #1: Dear Dr Nolte,

My major concern with the manuscript is regarding the translation and the cultural adaptation of the Patient Experience with Treatment and Self-Management (PETS) questionnaire.

A section in the manuscript titled “Adaptation of the PETS instrument” briefly mentions the steps undertaken by the team to translate the questions into Swahili.

But I feel it would be more appropriate to publish a full translational study of the Kenyan version initially elaborating on the process of translation and validation before this study.

Kind regards.

Reviewer #2: 1. Please do rationale of choosing two place where the study has been conducted.

2. How the sample size has been estimated? give the statistical tools used to estimate the sample

3. Please add PETS questionnaire in the supplementary.

4. Figures and not so visible, revise it.

Reviewer #3: The study evaluates an important perspective on patient centered research for NCDs. The authors have substantially highlighted the limitations of the study. Why this manuscript may be the first to utilise the approach stated in the methods, I am of the opinion that such a statement declaring "this is the first paper..." may be more subtly made. There could other similar research approaches that authors may not be aware of. Hence, if such declaration is deemed necessary, the word may could prefix the statement.

Reviewer #4: This is an interesting article that moves away from the assessment of disease burden and mortality to treatments that can become a significant burden in the management of chronic diseases for patients' quality of life.

It includes a validation component of PETS in the Kenyan context which was well described by the authors. But have the steps been documented? What was the level of concordance between translators at the different stages to better capture the challenge of the adaptation steps ? (I encourage the authors to publish this data in particular to help other teams to better prepare for this stage)

1. Can the participation rate be clarified ?

2. 60 minutes for a survey, especially by phone is not easy. How could fatigue have affected the quality of responses?

3. "The majority of respondents reported having at least one chronic condition" : as they were selected on the basis of having at least one chronic disease, 100% is expected. So can we delete that sentence or rephrase it?

4. The paragraph "Implications for practice and research" can be further clarified to clearly define the contribution of this work to change. Specify areas where investment needs to be strengthened for each of the communities and perhaps identify other factors that would be vulnerable to actions.

5. The conclusion (abstract and manuscript) should be more specific to describe the burden of treatment, the associated factors identified and open up on perceived actions as a response to improve the situation.

Reviewer #5: Comments:

1. The authors state that the objective of the paper was to explore treatment burden among people with NCDs in Kenya. Being stated this, the authors go on at length to explain the validation of the instrument (PETS), they have adapted to the Kenyan context. I am quite confused whether the authors want to present the validation of the instrument or the actual application of the survey instrument to assess the burden. Please explicitly state this in your objectives.

2. What was the rationale for choosing the particular instrument (PETS) to capture treatment burden?

3. In the participant recruitment section, I am not quite clear with the sample size calculations. The authors have used a SS of 300 and the rationale for this is stated as “this ss was used by the developers of the PETS survey instrument to test its validity”, but that cannot be the rationale here since you are using this sample to assess the treatment burden.

4. Why did the authors oversample? Did you expect non-response and how much was this percentage ?

5. How did the authors randomly select the patients from the study population?

6. The authors have stated they used the register as the sampling frame but patients from what period registered in the facilities were selected. And what if the randomly selected patient was not available or did not consent to the study.

7. In the analysis section, its not clearly stated the individual scoring of each item, and the minimum and maximum scoring. Why was the scale score transferred to 0 to 100 scale and what does 0 means and what does 100 means?

8. Can the results section first describe the validation if the authors want to include that also as the objective of the paper and then in the next section describe the finding of the survey using the validated instrument.

9. In the results section, suddenly the authors mention about 2 single item scale, why was this included? Was the validity of it checked? Can this be mentioned in the methodology.

6. PLOS authors have the option to publish the peer review history of their article (what does this mean?). If published, this will include your full peer review and any attached files.

**Do you want your identity to be public for this peer review?** For information about this choice, including consent withdrawal, please see our Privacy Policy.

Reviewer #1: No

Reviewer #2: **Yes: **Shubham Kumar

Reviewer #3: No

Reviewer #4: No

Reviewer #5: **Yes: **Sunu C Thomas

---

## [Editor Report · Decision Letter 1]

19 Dec 2022

Understanding the treatment burden of people with chronic conditions in Kenya: a cross-sectional analysis using the Patient Experience with Treatment and Self-Management (PETS) questionnaire

PGPH-D-22-01437R1

Dear Professor Nolte,

We are pleased to inform you that your manuscript 'Understanding the treatment burden of people with chronic conditions in Kenya: a cross-sectional analysis using the Patient Experience with Treatment and Self-Management (PETS) questionnaire' has been provisionally accepted for publication in PLOS Global Public Health.

Best regards,

Panniyammakal Jeemon

Academic Editor